# Tracing frequent users of regional care services using emergency medical services data: a networked approach

Laura Maruster ![ORCID],[1] Durk-Jouke van der Zee,[2] Jaap Hatenboer,[3] Erik Buskens[4]

¹Innovation Management & Strategy, Faculty of Economics and Business, University of Groningen, Groningen, The Netherlands
²Operations, Faculty of Economics and Business, University of Groningen, Groningen, The Netherlands
³Ambulancezorg, UMCG, Tynaarlo, Groningen, The Netherlands
⁴Epidemiology, University of Groningen, University Medical Center Groningen, Groningen, The Netherlands

**Correspondence to**
Dr Laura Maruster;
l.maruster@rug.nl

## ABSTRACT

**Objectives** This study shows how a networked approach relying on 'real-world' emergency medical services (EMS) records might contribute to tracing frequent users of care services on a regional scale. Their tracing is considered of importance for policy-makers and clinicians, since they represent a considerable workload and use of scarce resources. While existing approaches for data collection on frequent users tend to limit scope to individual or associated care providers, the proposed approach exploits the role of EMS as the network's 'ferryman' overseeing and recording patient calls made to an entire network of care providers.

**Design** A retrospective study was performed analysing 2012–2017 EMS calls in the province of Drenthe, the Netherlands. Using EMS data, benefits of the networked approach versus existing approaches are assessed by quantifying the number of frequent users and their associated calls for various categories of care providers. Main categories considered are hospitals, nursing homes and EMS.

**Setting** EMS in the province of Drenthe, the Netherlands, serving a population of 491 867.

**Participants** Analyses are based on secondary patient data from EMS records, entailing 212 967 transports and 126 758 patients, over 6 years (2012–2017).

**Results** Use of the networked approach for analysing calls made to hospitals in Drenthe resulted in a 20% average increase of frequent users traced. Extending the analysis by including hospitals outside Drenthe increased ascertainment by 28%. Extending to all categories of care providers, inside Drenthe, and subsequently, irrespective of their location, resulted in an average increase of 132% and 152% of frequent users identified, respectively.

**Conclusions** Many frequent users of care services are network users relying on multiple regional care providers, possibly representing inefficient use of scarce resources. Network users are effectively and efficiently traced by using EMS records offering high coverage of calls made to regional care providers.

## INTRODUCTION

Frequent users, that is, patients that make repetitive calls for healthcare services, may be responsible for a relatively large share of regional care consumption. They represent a minority of emergency department (ED) patients (4.5%–8%), yet, they may account for up to 21%–28% of all ED visits.[1–3] Different solutions have been devised for frequent users once identified. Subsequently, the appropriate answers to their needs, and consequently reducing the visits to ED and ambulance transports may be achieved. These solutions range from case management[4–6] to individual care plans,[7–9] and facilitated contacts with healthcare providers.[10] However, to be able to offer and consider such a form of advance care planning for apparently frail patients, they first need to be identified. The latter in reality may escape attention or appear difficult with data scattered over various institutions. Clearly, due to their high impact on care providers' workload and associated costs, they are a focal group for regional policy-makers and clinicians aiming to make the best use of scarce resources. In the Netherlands and possibly other settings, the emergency medical services (EMS) are increasingly overburdened, and at times encounter backlogs at the EDs of hospitals.[11 12] Indeed, the role of EMS in triage and adequate and timely referral is increasingly recognised in acute

## Strengths and limitations of this study

► Single-source comprehensive data such as emergency medical services (EMS) serving an entire region allow identification and tracing of frequent users of health services.
► EMS data include an extensive collection of patient transport data relating subsequent services provided by hospitals and nursing homes.
► A networked approach for tracing frequent users relying on EMS records allowed us to effectively and efficiently identify frequent users.
► We piloted EMS records for only one province in the Netherlands, while we expect that this generic approach could be easily transposed to any region.
► As EMS records only include frequent users who are not capable of self-transport, those frequent users who do not or rarely make use of EMS will not be traced by the proposed approach.

BMJ

care networks. Accordingly, identifying opportunities to relieve an overburdened acute care system from frequent and inappropriate may be considered an impending responsibility of EMS.

Notably, many frequent users appear to be network users, relying on multiple care providers.[13] Their choice of care providers is influenced by, for example, their preferences and care providers' specialisation. In particular, tracing patients' network use tends to be cumbersome. Hurdles not easily taken in data collection are, for example, rules on patient privacy, competition among care providers, incompatibility of information systems and efforts to be put in. Moreover, these hurdles likely imply high data collection costs. Not surprisingly, many research designs limit their scope to single or associated care providers, with a main focus on hospitals.[2 13–29] Hence, many factual frequent users may remain unnoticed.

Basically, current approaches towards data collection on frequent users stress probing of individual care providers.[23 30] Alternatively, acknowledging frequent users being network users, this article suggests a networked approach for their tracing, relying on EMS data. Acting as the 'ferryman' in the regional network, EMS oversee and record patient calls made to regional care providers, including hospitals and nursing homes. Importantly, the EMS patient population is likely to include many frequent users.[31 32] In addition, their need for mobile nursing services and transport indicates that their requirements of care resources may be high.

The aim of the present study is to show how the use of the proposed networked approach might efficiently contribute to tracing frequent users on a regional scale.

## METHODS
### Care network in the province of Drenthe
The province of Drenthe, the Netherlands, has a population of 491 867 inhabitants, with a population density of 183 inhabitants/km$^2$.[33] Hospital care for its population is provided by four hospitals within the province, and by several hospitals located in neighbouring provinces. Three of the hospitals in Drenthe offer basic treatment. In one hospital, the necessary skills and resources for treating multilevel traumas are present. Referral

to around 80 hospitals in other provinces is motivated by reasons such as their proximity to the patient scene, patient preferences, level of care or specialisation in specific treatments. Nursing care is provided by a few dozen of large homes, and around 300 smaller (specialised) homes, mainly located within the province. EMS are provided by a single operator, relying on a network of 14 bases in 13 cities/villages in Drenthe. Its services include both urgent and planned patient transports to hospitals and planned transports to nursing homes. Planned rides are legitimated by patient care needs that prohibit self-transport.

### Data
Patient data are collected from EMS records of ambulance rides performed between 1 January 2012 and 31 December 2017. Collected data include the rides' dates and times, and destinations, that is, care providers. EMS are marked as a formal care provider in case treatment provided by the ambulance nurse on scene suffices to address patient care needs, that is, EMS see & treat (EMS S&T), implying no involvement of other care providers. Motivated by EMS scope of services, three categories of care providers are distinguished, that is, hospitals, nursing homes and EMS S&T.

### Privacy and approval
Since the data are routinely collected for administrative purposes, and completely anonymised, that is, there is no direct contact with identifiable persons, this study does not fall within the scope of the Medical Research Involving Human Subjects Act (Wet Maatschappelijke Ondersteuning).[34] We obtained a full waiver for using anonymised data from the EMS services from our institutional ethical review board.

### Patient and public involvement
No patient and public involved.

### Data analysis
For data analysis, EMS records referring to single rides are anonymised, cleaned by removing empty records, that is, records not relating to patients, and inspected for correctness of data provided. Patients qualify as a frequent user

**Table 1** Number of frequent users, using data on care provider categories, that is, hospitals, nursing homes, emergency medical services (EMS) see and treat (S&T) and all care providers, located in Drenthe

| Year | Hospitals—no data sharing | Hospitals | Nursing homes | EMS S&T | All care providers | All care providers/hospitals—no data shared (%) |
|------|---------------------------|-----------|---------------|---------|--------------------|-------------------------------------------------|
| 2012 | 189 | 222 | 34 | 15 | 398 | 211 |
| 2013 | 153 | 181 | 42 | 16 | 340 | 222 |
| 2014 | 204 | 245 | 22 | 19 | 495 | 243 |
| 2015 | 253 | 309 | 18 | 68 | 635 | 251 |
| 2016 | 279 | 321 | 28 | 46 | 611 | 219 |
| 2017 | 263 | 332 | 30 | 33 | 649 | 247 |

**Table 2** Number of calls corresponding to frequent users, using data on care provider categories, that is, hospitals, nursing homes, emergency medical services (EMS) see and treat (S&T) and all care providers, located in Drenthe

| Year | Hospitals—no data sharing | Hospitals | Nursing homes | EMS S&T | All care providers | All care providers/ hospitals—no data shared (%) |
|------|---------------------------|-----------|---------------|---------|--------------------|--------------------------------------------------|
| 2012 | 1161 | 1296 | 283 | 84 | 2423 | 209 |
| 2013 | 1158 | 1279 | 497 | 73 | 2503 | 216 |
| 2014 | 1386 | 1557 | 431 | 106 | 3204 | 231 |
| 2015 | 1477 | 1711 | 174 | 388 | 3597 | 244 |
| 2016 | 1772 | 1955 | 229 | 245 | 3631 | 205 |
| 2017 | 1536 | 1821 | 193 | 170 | 3581 | 233 |

if they meet a threshold of four calls in a calendar year. Although definitions differ, usually a threshold of four to five calls or more per year is used to classify a patient as a 'frequent user'.[35 36] Frequent users are quantified by presenting their numbers and number of calls, including yearly trend figures.

The potential of the proposed network-based approach for data collection on frequent users is evaluated by assessing its benefits compared with existing approaches. Whereas the proposed approach relies on EMS data, existing approaches build on data obtained from individual care providers. In principle, both approaches may render similar outcomes. However, existing approaches face hurdles not easily overcome, due to the fact that multiple organisations, that is, care providers, are involved in data collection. Known hurdles are rules on patient privacy, competition among care providers, incompatibility of information systems, efforts to be put in and costs of overcoming hurdles. They likely restrict the scope of data collection, that is, the number of care providers being considered. Restriction of scope may affect identifying patients making calls to various care providers as frequent users after combining and quantifying their calls. The proposed approach relies on a single source of data, and does overcome these scoping decisions.

Effects of the choice of scope on the number of frequent users identified and their associated calls are studied by considering alternative subsets of EMS records. Choice of subsets is related to provider categories, that is, hospitals, nursing homes and/or EMS S&T, and their location, that is, inside or outside Drenthe. By either allowing patient records to be combined for chosen subsets of care providers, or not, beneficial effects of the possibility to identify network users are assessed. This effect is studied for hospitals, serving most of the patient calls.

## RESULTS

EMS records for 2012–2017 refer to 212 967 calls for services, involving 126 758 patients. Data cleaning resulted in 2494 calls being removed. In addition, 13 156 calls (6%) were discarded due to unknown, not recorded destinations (ie, care providers). The remaining 199 811 calls are included in the study. Out of these 199 811 calls, 147 027 (74%), 10 976 (5%) and 41 808 (21%) refer to services provided by hospitals, nursing homes and EMS S&T, respectively. Results of the evaluation of the networked approach for data collection on frequent users are shown in tables 1–4. Tables 1 and 2 quantify the number of *frequent users* and their associated *calls* for alternative choices of categories of care providers located in Drenthe on a yearly basis. Categories of care providers considered are hospitals, nursing homes, EMS S&T and all care providers, that is, taking all aforementioned categories together. Except for hospitals, all results assume data sharing among care providers within categories set,

**Table 3** Number of frequent users, data on care provider categories, that is, hospitals, nursing homes, emergency medical services (EMS) see and treat (S&T) and all care providers, located in and outside Drenthe

| Year | Hospitals—no data sharing | Hospitals | Nursing homes | EMS S&T | All care providers | All care providers/ hospitals—no data sharing (%) |
|------|---------------------------|-----------|---------------|---------|--------------------|---------------------------------------------------|
| 2012 | 256 | 368 | 35 | 15 | 578 | 226 |
| 2013 | 204 | 285 | 44 | 16 | 486 | 238 |
| 2014 | 261 | 395 | 25 | 19 | 706 | 270 |
| 2015 | 308 | 443 | 21 | 72 | 825 | 268 |
| 2016 | 344 | 511 | 30 | 47 | 845 | 246 |
| 2017 | 330 | 531 | 32 | 33 | 881 | 267 |

Table 4 Number of calls corresponding to frequent users, data on care provider categories, that is, hospitals, nursing homes, emergency medical services (EMS) see and treat (S&T) and all care providers, located in and outside Drenthe

| Year | Hospitals—no data sharing | Hospitals | Nursing homes | EMS S&T | All care providers | All care providers/ hospitals—no data sharing (%) |
|---|---|---|---|---|---|---|
| 2012 | 1984 | 2468 | 287 | 85 | 3826 | 193 |
| 2013 | 1829 | 2180 | 506 | 73 | 3658 | 200 |
| 2014 | 2120 | 2699 | 444 | 106 | 4685 | 221 |
| 2015 | 2116 | 2693 | 194 | 404 | 4902 | 232 |
| 2016 | 2515 | 3245 | 239 | 250 | 5228 | 208 |
| 2017 | 2220 | 3082 | 210 | 171 | 5133 | 231 |

allowing frequent users being network users to be traced. In addition, hospitals results are shown for settings where such data sharing among single hospitals is not possible. Hence, network users may be neglected. For respective settings, the number of unique frequent users is shown, that is, numbers are corrected for the fact that a single patient may be classified as a frequent user for multiple hospitals. The final column indicates the effect of combining data for all care providers versus a setting where frequent users of hospital services are identified by studying single hospitals in isolation. It shows how the number of frequent users traced and their associated calls increase by more than a twofold by combining data for all care providers. Similar to tables 1–4 quantify the annual number of *frequent users* and their associated *calls* for alternative choices of categories of care providers, without setting requirements to their location. Final columns in tables 3 and 4 show which numbers of frequent users and their associated calls are found when using the full EMS data set, including care providers located outside Drenthe. These amount to around 2.5 times the numbers found when studying hospitals in isolation.

## DISCUSSION

Tracing frequent users, that is, patients displaying a high consumption (instead of appeal here and elsewhere) of health services, is considered highly relevant in regional policy-making. This is due to their high impact on care provider workload and use of scarce resources. The results of this study demonstrate that a networked approach for tracing frequent users relying on EMS data is capable of effectively and efficiently identifying frequent users. Case-related results for the province of Drenthe indicate how more than a twofold frequent users may be traced by the proposed approach relative to existing approaches, relying on data collection by questioning individual care providers. Moreover, these results are obtained using a single source of data, whereas existing approaches would have required questioning a few hundred care providers.

Success of the proposed approach builds on its scope. As a straightforward effect of including more care providers, that is, hospitals, nursing homes and EMS, located in and outside the region, more frequent users are traced and

more accurately so. Most gains result from the possibility of combining data from different providers thus tracing those frequent users being network users, that is, making use of multiple care providers, possibly representing inefficient use of scarce resources. Network users may easily be overlooked in existing approaches due to restrictions on their reach, following from, for example, competition among care providers involved, incompatible information systems and efforts to be put in data collection involving many providers. Case-related results for the province of Drenthe indicate on average a 20% and 13% increase of the number of frequent users identified and their associated calls traced over the observation period, if hospital data were combined. Including hospitals outside Drenthe in this analysis improved the identification by another 28% and 52% on average. Extending scope to all categories of care providers, first focusing only on the Drenthe location, and second, setting no restrictions on care providers location, results in an increase of around 132% and 152% of frequent users identified. At the same time, their associated calls increase by 123% and 114%, respectively. Gains found are relatively constant over the observation period. Relevance of being able to trace this group of network users follows from its expected growth among others resulting from ongoing specialisation in Dutch healthcare and outside. Moreover, their existence and upsurge may call for increased regional coordination among care providers to safeguard care continuity and avoid fragmented care and wrong referrals.[20] Clearly, being aware of frequent users is paramount to undertaking appropriate action. The opportunity we identified and seized might seem trivial in settings where individuals are easily traced, that is, single payer or service provider systems. In these systems, the necessity to take appropriate action is no less urgent, yet the effort to obtain a listing and pattern of use might be simpler. Nevertheless, we provide a worked out exemplary approach that may be applied in many settings like the Netherlands.

The proposed networked approach may serve as a stepping stone in analysing consumption patterns of frequent users on a regional scale. Once frequent users have been identified by the approach, techniques such as process and data mining may allow for further groupwise

analysis of patients routings along care providers, and their (joint) care needs following from EMS diagnostic data. These techniques have been successfully used to analyse healthcare processes, usually in an intrahospital context.[37–40] Whereas process mining may be helpful in capturing patients' routing along care providers, data mining may assist in analysing patients' care needs further using text analysis of diagnostic data, thereby unravelling their reasons for calls. Consumption patterns thus revealed may refine insights among policy-makers on frequent users care needs, and their use of care services. In turn, revealing unfamiliar or much traversed patient routings may be helpful in, for example, optimising these by concerting activities among care providers or evoking patient treatment plans, thus improving and safeguarding quality of care.

The present study has limitations. First, only EMS records for the province of Drenthe, the Netherlands, are studied. Clearly, regional characteristics may have an effect on the results of the proposed approach. However, while this may be true, its success is not expected to be dependent on location or region, but relies on EMS' role as the regional ferryman and its records that oversee patient calls for service to a great many care providers. Second, EMS records only include frequent users who are not capable of self-transport. Thus, frequent users who do not, or rarely, make use of EMS will not be traced by the proposed approach. Third, the success of any approach depends on the quality of the underlying data. We found how inclusion of ambulance transports to unknown, that is, not recorded destinations in data analysis may result in higher numbers of frequent users and their associated calls being identified. Fourth, as it is explorative, the paper signifies the potential of the proposed approach for tracing frequent users and enhancing regional policy-making. Ongoing and future research should be directed towards methodological issues concerning the use of the approach and its tradeoff with alternative approaches.

## CONCLUSIONS

Many frequent users of care services are network users relying on multiple regional care providers, possibly representing inefficient use of scarce resources. Network users are effectively and efficiently traced by using EMS records offering high coverage of calls made to regional care providers.

**Acknowledgements** The authors would like to thank Harriëtte Holt from UMCG Ambulancezorg for providing input on the manuscript as domain expert, and Coert Schrijver for his technical support.

**Contributors** All authors contributed to the conception and design of the study, read and approved the submitted manuscript. LM contributed to data collection. LM and D-JvdZ contributed to data analysis, interpretation of the data and drafting the manuscript. JH and EB contributed to data analysis, interpretation of the data and to the revision of the paper.

**Funding** The authors have not declared a specific grant for this research from any funding agency in the public, commercial or not-for-profit sectors.

**Competing interests** None declared.

**Patient consent for publication** Not required.

**Provenance and peer review** Not commissioned; externally peer reviewed.

**Data availability statement** No additional data are available.

**ORCID iD**
Laura Maruster http://orcid.org/0000-0002-6588-7648

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
