## [Reviewer comments · BMJ Open]

ARTICLE DETAILS

TITLE (PROVISIONAL)	Tracing frequent users of regional care services using emergency medical services data – a networked approach
AUTHORS	Maruster, Laura; van der Zee, Durk-Jouke; Hatenboer, Jaap; Buskens, Erik

VERSION 1 – REVIEW

REVIEWER	Dr Di Mauro Raffaele Emergency Department - Foundation IRCCS Ca' Granda Ospedale Maggiore Policlinico of Milan, Italy
REVIEW RETURNED	18-Dec-2019

GENERAL COMMENTS	The study appears to be easy to interpret and reproduce. Methods are linear and appropriate. The introduction should be complemented by better clarifying the appropriate way in which EDs manage frequent users with regard to current scientific recommendations. It would be recommended to add more bibliography references. Frequent users, i.e., patients that make repetitive calls for health care services, may be responsible for a relatively large share of regional care consumption. They represent a minority of Emergency Department (ED) patients (4.5-8%), yet, they may account for up to 21-28% of all ED visits 1. Many studies have shown that a customized Case Management approach helps Frequent Users in finding an appropriate answer to their needs reducing visits to the ED and, in some cases, healthcare costs through Individual Care Plans, telephone contact and facilitated contacts with healthcare providers. Due to their high impact on care providers' workload and associated costs they are a focal group for regional policy makers and clinicians aiming to make best use of scarce resources.
--

REVIEWER	Joonghee Kim Seoul National University Bundang Hospital
REVIEW RETURNED	16-Feb-2020

GENERAL COMMENTS	The authors showed EMS dataset (ambulance call log data) can be used to identify heavy users of health care resources. The authors concluded this is a better approach compared to those depending on single facility dataset. I think this is so obvious and not need to be proved by any research. In addition, there are already many previous studies providing deeper insights on this subjects which were not mentioned in this paper.
---

REVIEWER	Yohann Chiu Université de Sherbrooke, Canada
REVIEW RETURNED	19-Feb-2020

GENERAL COMMENTS	This is an interesting piece of work about tracing frequent users of care services through Emergency medical services, in the province of Drenthe. It is well written and the results are very encouraging. I have a few comments, though I suspect that they are mostly minor. 1) EMS should appear in full at first mention in the text. It is in full mention in the abstract, but I think that it also should be written in full at page 5 (line 45). In the same spirit, Netherlands should appear at first mention of the province of Drenthe (page 6 line 8). 2) How did the authors gain access to EMS data? Since there is no detail in the Data or Privacy sections, it is not clear to me as a non Deutch researcher how easy the access would be. As the authors state themselves, there are many hurdles in gaining access to those data, so I am curious as to how they . 3) This is related to my previous comment; more globally, has any research on frequent use been done using those same databases? The results suggest that using EMS data allows for tracing a lot more frequent users than any single care providers. This seems a very good improvement and, depending on the answer to my previous question, I was wondering what other studies have shown using those databases. 4) “We found how inclusion of ambulance transports to unknown, i.e., not recorded destinations in data analysis may result in higher numbers of frequent users” (page 12 line 28). What proportion of ambulance transports was related to unknown destinations? I did not see any mention of this in the text. 5) In order to get a full grasp on the results scope, I would be interested in seeing which type of frequent users are “captured” using EMS databases (e.g. patients with drug abuse, geriatric patient with comorbidity, etc.). Are the reason for EMS calls included in the databases? If so, would it be possible to investigate those reasons, or at least to discuss it in the discussion?
--

VERSION 1 – AUTHOR RESPONSE

Reviewer: 1: Dr Di Mauro Raffaele
Emergency Department - Foundation IRCCS Ca' Granda Ospedale Maggiore Policlinico of Milan, Italy

The study appears to be easy to interpret and reproduce. Methods are linear and appropriate.

Answer:

Thank you for giving us this opportunity to revise our paper. The constructive comments indeed guided us in making relevant changes to the manuscript.

1. The introduction should be complemented by better clarifying the appropriate way in which EDs manage frequent users with regard to current scientific recommendations. It would be recommended to add more bibliography references.

“Frequent users, i.e., patients that make repetitive calls for health care services, may be responsible for a relatively large share of regional care consumption. They represent a minority of Emergency Department (ED) patients (4.5-8%), yet, they may account for up to 21-28% of all ED visits.^{1 2}”

Many studies have shown that a customized Case Management approach helps Frequent Users in finding an appropriate answer to their needs reducing visits to the ED and, in some cases, healthcare costs through Individual Care Plans, telephone contact and facilitated contacts with healthcare providers.

“Due to their high impact on care providers’ workload and associated costs they are a focal group for regional policy makers and clinicians aiming to make best use of scarce resources.”

Answer:

We thank the reviewer for providing us with recommendations to include more references regarding the management of frequent users, especially by a case management approach, individual care plans, telephone contact and facilitated contacts with healthcare providers. While these solutions might indeed reduce unnecessary use of services it also remains clear that not all individuals can be identified prospectively, i.e., before such individuals become frail and frequent users, that would profit from some sort of advance care planning. We propose that exploiting routinely collected data as described would be complementary, and could allow factual identification and subsequently taking appropriate measures as suggested.

Accordingly, new sentences were added (see lines 74-80 in paper, and text below), including additional references:

“Different solutions have been devised for frequent users once identified. Subsequently, the appropriate answers to their needs, and consequently reducing the visits to ED and ambulance transports may be achieved. These solutions range from case management⁴⁻⁶, to individual care plans⁷⁻⁹, and facilitated contacts with healthcare providers¹⁰. However, to be able to offer and consider such a form of advance care planning for apparently frail patients they first need to be identified. The latter in reality may escape attention or appear difficult with data scattered over various institutions.”

4. Hudon C, Chouinard MC, Pluye P, et al. Characteristics of Case Management in Primary Care Associated With Positive Outcomes for Frequent Users of Health Care: A Systematic Review. *Annals of Family Medicine* 2019;17(5):448-58.

5. Moschetti K, Iglesias K, Baggio S, et al. Health care costs of case management for frequent users of the emergency department: Hospital and insurance perspectives. *Plos One* 2018;13(9)

6. Grover CA, Sughair J, Stoopes S, et al. Case Management Reduces Length of Stay, Charges, and Testing in Emergency Department Frequent Users. *Western Journal of Emergency Medicine* 2018;19(2):238-44.

7. Moe J, Kirkland SW, Rawe E, et al. Effectiveness of Interventions to Decrease Emergency Department Visits by Adult Frequent Users: A Systematic Review. *Academic Emergency Medicine* 2017;24(1):40-52.

8. Pope D, Fernandes CMB, Bouthillette F, et al. Frequent users of the emergency department: a program to improve care and reduce visits. *Canadian Medical Association Journal* 2000;162(7):1017-20.

9. Spillane LL, Lumb EW, Cobaugh DJ, et al. Frequent users of the emergency department: Can we intervene? *Academic Emergency Medicine* 1997;4(6):574-80.
10. Agarwal G, Pirrie M, McLeod B, et al. Rationale and methods of an Evaluation of the Effectiveness of the Community Paramedicine at Home (CP@home) program for frequent users of emergency medical services in multiple Ontario regions: a study protocol for a randomized controlled trial. *Trials* 2019;20.

In addition, we added two references to underpin the relative share of frequent users in ED inflow (see line 73):

1. Hudon C, Courteau J, Krieg C, et al. Factors associated with chronic frequent emergency department utilization in a population with diabetes living in metropolitan areas: a population-based retrospective cohort study. *BMC Health Serv Res* 2017;17(1):525.
2. Doupe MB, Palatnick W, Day S, et al. Frequent Users of Emergency Departments: Developing Standard Definitions and Defining Prominent Risk Factors. *Annals of Emergency Medicine* 2012;60(1):24-32.

Also, to stress frequent users being network users, we added another reference (see line 88):

13. Castillo EM, Brennan JJ, Killeen JP, et al. IDENTIFYING FREQUENT USERS OF EMERGENCY DEPARTMENT RESOURCES. *Journal of Emergency Medicine* 2014;47(3):343-47.

Reviewer: 2: Joonghee Kim
Seoul National University Bundang Hospital

The authors showed EMS dataset (ambulance call log data) can be used to identify heavy users of health care resources. The authors concluded this is a better approach compared to those depending on single facility dataset. I think this is so obvious and not need to be proved by any research. In addition, there are already many previous studies providing deeper insights on this subjects which were not mentioned in this paper.

Answer:

Part of the remark put forward by the reviewer is true, i.e., a single comprehensive record of some sort capturing all healthcare use of individuals indeed would allow simple and efficient identification of heavy users. However, only those countries or regions that have a single payer or single service providing organization indeed have those possibilities. In most settings, funding as well as service provision is dispersed and impossible to completely trace. This study presents an exemplar of an inner service provider, i.e., EMS that links the majority of acute and chronic health services and thus is in a unique position that other service providers within the network will never achieve. We maintain that for many regions and settings this insight is novel and worthwhile to consider. Indeed, we respectfully disagree with the reviewer. Importantly, also the third reviewer working in a setting more comparable to ours also recognizes the relevance and novelty.

We also added the following sentences in section "Discussion", see lines 239-243:

"The opportunity we identified and seized might seem trivial in settings where individuals are easily traced, i.e., single payer or service provider systems. In these systems the necessity to take appropriate action is no less urgent, yet the effort to obtain a listing and pattern of use might be simpler. Nevertheless, we provide a worked out exemplary approach that may be applied in many settings like the Netherlands."

Reviewer: 3

Reviewer Name: Yohann Chiu

Université de Sherbrooke, Canada

This is an interesting piece of work about tracing frequent users of care services through Emergency medical services, in the province of Drenthe. It is well written and the results are very encouraging. I have a few comments, though I suspect that they are mostly minor.

Answer:

Thank you for giving us this opportunity to revise our paper. The constructive comments indeed guided us throughout all the changes we made in this revised version.

1. EMS should appear in full at first mention in the text. It is in full mention in the abstract, but I think that it also should be written in full at page 5 (line 45). In the same spirit, Netherlands should appear at first mention of the province of Drenthe (page 6 line 8).

Answer:

Thank you for helping us improving our manuscript, we incorporated the suggestions on EMS (see lines 57 and 82-83). Also we have inserted "the Netherlands", following reviewer's suggestion (see line 109).

2. How did the authors gain access to EMS data? Since there is no detail in the Data or Privacy sections, it is not clear to me as a non Dutch researcher how easy the access would be. As the authors state themselves, there are many hurdles in gaining access to those data, so I am curious as to how they .

Answer:

Thank you for this question. We were able to overcome the mentioned hurdles by getting a strong commitment from the EMS provider serving the Province of Drenthe, the Netherlands. The EMS services play a key role in our regional emergency care networks, and they were at times experiencing congestion. They thus were very much interested in analyzing their data and looking for causes and potential solutions. Indeed, they may be considered problem owners looking for expert partners such as our team to collaborate with. We obviously also obtained a full waiver for using anonymized data from the EMS services from our institutional ethical review board as stated in lines 132-134 (see also below):

"We obtained a full waiver for using anonymized data from the EMS services from our institutional ethical review board."

Furthermore, to clarify the interest of the EMS in identifying frequent users and cooperating with us we add the following lines to the Introduction, see lines 82-86.

"In the Netherlands and possibly other settings the Emergency Medical Services (EMS) are increasingly overburdened, and at times encounter backlogs at the EDs of hospitals^{11 12} Indeed, the role of EMS in triage and adequate and timely referral is increasingly recognised in acute care networks. Accordingly, identifying opportunities to relieve an overburdened acute care system from frequent and inappropriate may be considered an impending responsibility of EMS."

3. This is related to my previous comment; more globally, has any research on frequent use been done using those same databases? The results suggest that using EMS data allows for tracing a lot more frequent users than any single care providers. This seems a very good improvement and, depending on the answer to my previous question, I was wondering what other studies have shown using those databases.

Answer:

To the best of our knowledge, this is the first manuscript addressing the use of EMS data as key and unique data source for tracing frequent users.

4. "We found how inclusion of ambulance transports to unknown, i.e., not recorded destinations in data analysis may result in higher numbers of frequent users" (page 12 line 28). What proportion of ambulance transports was related to unknown destinations? I did not see any mention of this in the text.

Answer:

In section "Results" (lines 165-167), we explain which data is used in analysis:

"EMS records for 2012-2017, refer to 212,967 calls for services, involving 126,758 patients. Data cleaning resulted in 2,494 calls being removed. In addition, 13,156 calls were discarded due to their lack of information on transport destination, i.e., care providers. The latter 13,156 calls refer to unknown destinations."

To clarify matters, we adapted the above text using same terminology as being used on the first paragraph of section "Results", see lines 165-167.

"EMS records for 2012-2017, refer to 212,967 calls for services, involving 126,758 patients. Data cleaning resulted in 2,494 calls being removed. In addition, 13,156 calls (6%) were discarded due to unknown, not recorded destinations (i.e. care providers)."

5. In order to get a full grasp on the results scope, I would be interested in seeing which type of frequent users are "captured" using EMS databases (e.g. patients with drug abuse, geriatric patient with comorbidity, etc.). Are the reason for EMS calls included in the databases? If so, would it be possible to investigate those reasons, or at least to discuss it in the discussion?

Answer:

We thank the reviewer for this excellent and very relevant remark.

EMS data include reasons for calls building on diagnostics provided by the ambulance nurse and 911 (Europe 112).

We added the following text in "Discussion", see lines 250-252 to clarify how reasons for EMS calls may be determined using novel techniques for diagnostics data analysis:

"Whereas process mining may be helpful in capturing patients' routing along care providers, data mining may assist in analysing patients' care needs further using text analysis of diagnostic data, thereby unravelling their reasons for calls."

Our ongoing work focuses on interpreting EMS diagnostic data using aforementioned techniques for identifying patient groups among frequent users, being in need of better service and/or possibly using scarce resources inefficiently.

VERSION 2 – REVIEW

REVIEWER	Dr Raffaele Di Mauro Fondazione IRCCS Ca' Granda - Ospedale Maggiore Policlinico, Milan (Italy)
REVIEW RETURNED	06-Apr-2020

GENERAL COMMENTS	I recommend reading this article, it could be of your interest to better clarify the case management activities that improve patient management in the emergency department. best regards
--

REVIEWER	Yohann Chiu Université de Sherbrooke, Canada
REVIEW RETURNED	24-Apr-2020

GENERAL COMMENTS	The authors answered to all the reviewers' comments and made appropriate changes in the manuscript.
---